# Molecular basis for redox control by the human cystine/glutamate antiporter system xc⁻

Joanne L. Parker 🆔 [1,6✉], Justin C. Deme 🆔 [2,3,4,6], Dimitrios Kolokouris 🆔 [1], Gabriel Kuteyi[1], Philip C. Biggin 🆔 [1], Susan M. Lea 🆔 [2,3,4✉] & Simon Newstead 🆔 [1,5✉]

Cysteine plays an essential role in cellular redox homoeostasis as a key constituent of the tripeptide glutathione (GSH). A rate limiting step in cellular GSH synthesis is the availability of cysteine. However, circulating cysteine exists in the blood as the oxidised di-peptide cystine, requiring specialised transport systems for its import into the cell. System xc⁻ is a dedicated cystine transporter, importing cystine in exchange for intracellular glutamate. To counteract elevated levels of reactive oxygen species in cancerous cells system xc⁻ is frequently upregulated, making it an attractive target for anticancer therapies. However, the molecular basis for ligand recognition remains elusive, hampering efforts to specifically target this transport system. Here we present the cryo-EM structure of system xc⁻ in both the apo and glutamate bound states. Structural comparisons reveal an allosteric mechanism for ligand discrimination, supported by molecular dynamics and cell-based assays, establishing a mechanism for cystine transport in human cells.

[1] Department of Biochemistry, University of Oxford, Oxford OX1 3QU, UK. [2] Dunn School of Pathology, University of Oxford, Oxford OX1 3RE, UK. [3] Central Oxford Structural Molecular Imaging Centre, University of Oxford, South Parks Road, Oxford OX1 3RE, UK. [4] Center for Structural Biology, Center for Cancer Research, National Cancer Institute, Frederick, MD 21702, USA. [5] The Kavli Institute for Nanoscience Discovery, University of Oxford, Oxford OX1 3QU, UK. [6]These authors contributed equally: Joanne L. Parker, Justin C. Deme. ✉email: joanne.parker@bioch.ox.ac.uk; susan.lea@nih.gov; simon.newstead@bioch.ox.ac.uk

Elevated levels of reactive oxygen species (ROS) play fundamental roles in many aspects of tumour development[1,2] and autoimmune diseases[3]. Increased levels of cytoplasmic ROS are generated through increased metabolic activity, mitochondrial dysfunction and oncogene activity which drive critical signalling pathways to sustain hyperproliferative growth[4] and inflammation[3]. More recently, regulation of ROS levels in cancer cells have been identified as an important factor in reducing ferroptosis, a nonapoptotic pathway linked to iron-dependent oxidative cell death[5,6]. The glutathione system plays a central role in redox homoeostasis and consists of glutathione (GSH), glutathione reductase, glutathione peroxidase and glutathione-S-transferase[7]. A key step in GSH synthesis is the availability of cysteine, which together with glutamate and glycine form the constituent parts of the tripeptide[8]. Free cysteine in the blood exists in its oxidised form cystine, wherein two thiol groups form a disulphide bond to create a unique dipeptide molecule that exhibits enhanced stability over the single amino acid. Under redox stress conditions mammalian cells use a specific system to increase cystine uptake to increase GSH synthesis, termed system xc⁻ a dedicated cystine-glutamate antiporter (Fig. 1a)[9], which under normal physiological conditions is predominantly expressed in astrocytes within the brain and macrophages[10,11]. However, system xc⁻ is upregulated in many different cancers[12], where expression has been linked to chemotherapeutic resistance[13] and poor survival in patients with glioblastoma[14].

The development of specific system xc⁻ inhibitors has therefore emerged as a promising route to supress tumour growth[15,16].

System xc⁻ belongs to the SoLute Carrier 7 (SLC7) family of secondary active amino acid transporters and is a heterodimer, composed of a light chain (also called xCT or SLC7A11) and a heavy chain 4F2hc. The heavy chain 4F2hc is a single transmembrane spanning glycoprotein (also called CD98hc or SLC3A2)[17,18] necessary for correctly trafficking the transporter to the plasma membrane[18,19]. Although several diverse functions have been linked to 4F2hc, including integrin signalling and cell proliferation[20,21], the role of the heavy chain on transporter function is less well understood. The SLC7 family regulates the flow of amino acids in the body[22,23] and can be divided into two subgroups, the cationic amino acid transporters (CATs, SLC7A1–4 and SLC7A14)[24] and the L-type amino acid transporters, or LATs (SLC7A5-13 and SLC7A15)[25], to which system xc⁻ belongs.

Recent high resolution cryogenic electron microscopy (cryo-EM) structurers of L-type amino acid transporters LAT1/4F2hc, LAT2/4F2hc and b[0,+]AT/rBAT revealed the structural organisation of the heterodimer[26–31], with the large glycosylated ectodomain sitting atop the transporter. The LAT subunits are members of the amino acid polyamine cation (APC) superfamily[32], which exhibit a conserved '5 + 5 inverted topology' fold, wherein the first five TM helices are related to the second five helices via a pseudo two-fold symmetry axis running parallel to the plane of

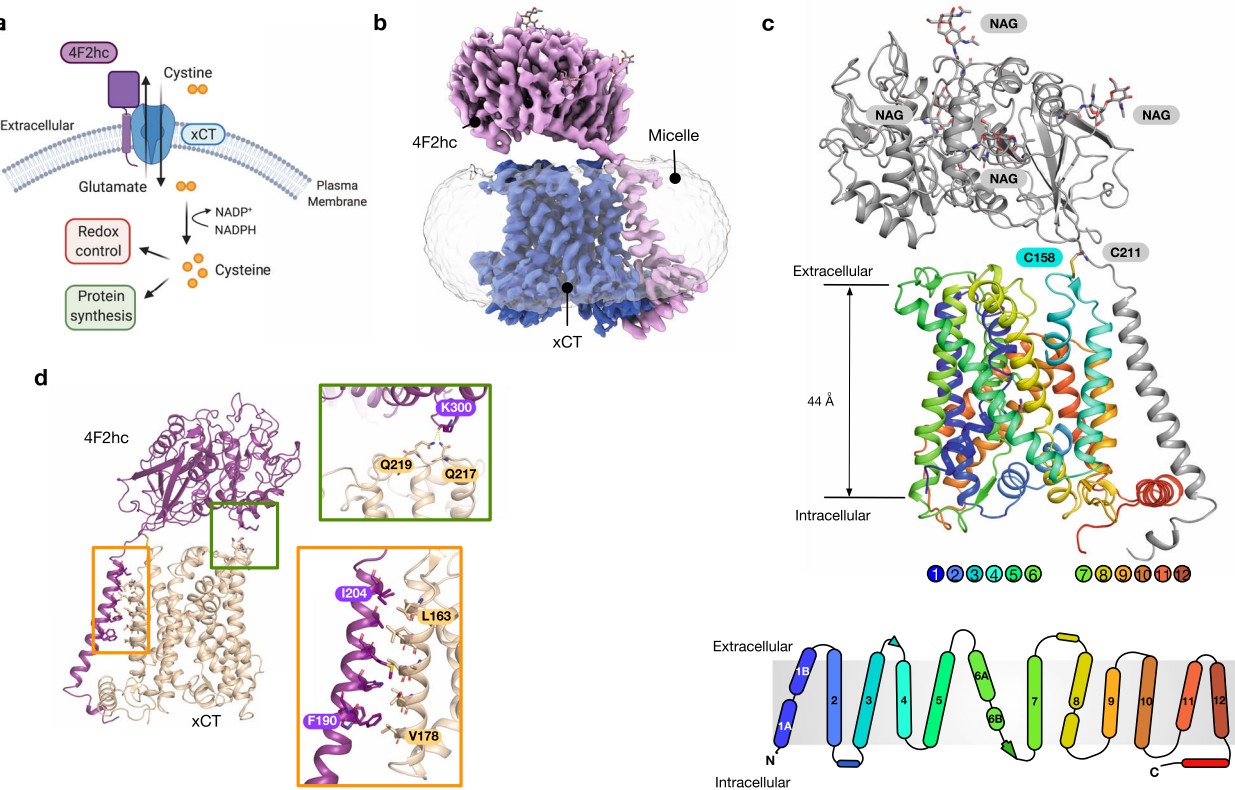

**Fig. 1 Cryo-EM structure of system xc⁻. a** System xc⁻ consists of a heavy chain component, 4F2hc (SLC3A2) and a transport module, xCT (SLC7A11), and functions as a dedicated cystine import system, exchanging extracellular L-cystine for intracellular L-glutamate. Following L-cystine transport into the cell, the molecule is rapidly reduced to L-cysteine via cystine reductase and used to regulate cellular redox levels via glutathione, alternatively free cysteine can also enter the protein synthesis pathway. **b** Cryo-EM density of 4F2hc (magenta) and xCT (blue) contoured at a threshold level of 1.0, superposed with lower-contoured threshold level of 0.22 to display the detergent micelle (grey). **c** Cartoon representation of system xc⁻, with 4F2hc (grey) showing conserved N-linked glycans (NAG). xCT is coloured blue from the N-termini to red at the C-terminus. The cysteines forming the disulphide which connects the two components are also indicated. A topology diagram is also shown of SLC7A11, using the same colours as in the main panel. **d** Analysis of the interactions between 4F2hc and xCT. The zoomed in views show the interactions between the extracellular domain (green box) and transmembrane domains (orange box).

the membrane[33]. Although structures of prokaryotic homologues of LATs revealed key features of the ligand binding site[22,34,35] many fundamental questions concerning amino acid recognition, selectivity and transport mechanism remain. To address these questions for cystine transport, we determined the cryo-EM structure of the human system xc⁻ in both the apo and L-glutamate bound states. Using a combination of cell-based assays and molecular dynamics we identified the key structural features of the binding site that enables glutamate and cystine recognition, revealing a previously unobserved mechanism for ligand discrimination. Our results provide valuable insights into cystine transport and establish a structural platform for understanding the role of xCT in redox homoeostasis in the cell.

## Results

**Cryo-EM structure of human system xc⁻.** Human SLC transporters present several challenges to recombinant overexpression and subsequent purification and analysis. Following several rounds of optimisation, we determined the optimal expression conditions for the xCT/4F2hc heterodimer using suspension HEK293 cells and dual affinity tags for selective capture of correctly folded protein (Supplementary Fig. 1). Following sample optimisation in LMNG-CHS, the apo structure was determined using single particle cryo-EM to an overall resolution of 3.4 Å (Fig. 1b, Table 1 & Supplementary Fig. 2), extending the resolution of a thermostabilised, transport deficient variant of xCT reported at 6.2 Å[31].

The light subunit of human xc⁻ transporter, xCT, (SLC7A11), consists of 12 transmembrane helices (TMs) (Fig. 1c and Supplementary Fig. 3), which adopt the canonical APC superfamily fold, wherein TM1-TM5 are related to TM6-TM10 by a pseudo two-fold symmetry axis located in the plane of the membrane[36]. The heavy chain of the transporter, 4F2hc (SLC3A2), which is also shared with LAT1 (SLC7A5) and LAT2 (SLC7A8), was also resolved in the map, including the disulphide bond between Cys211 in 4F2hc and Cys158 in xCT and the four N-glycosylation sites. The structure of the 4F2hc ectodomain is similar to that obtained in LAT1, LAT2 and the lower resolution xc⁻ structure[28,29,31,37]. As reported for LAT1, the single TM helix in 4F2hc interacts extensively with TM4, predominantly through hydrophobic interactions and 'knobs into holes' packing, with additional interactions observed with the C-terminal lateral helix in SLC7A11 (Fig. 1d). However, the position of the ectodomain relative to the transporter is noticeably different compared to the positions in LAT1 and LAT2, being elevated ~ 20° vertically away from the entrance to the binding site (Supplementary Fig. 4). This position is stabilised through an interaction between the backbone carbonyl of Lys300 on 4F2hc and, Gln217 and Gln219 in SLC7A11 (Fig. 1d). The repositioning of the ectodomain in our structures results in a wider entrance vestibule, possibly to accommodate the larger cystine ligand. SLC7A11 also has a noticeably less charged tunnel leading to the extracellular gate compared to LAT1 and LAT2, which may also facilitate binding of the cystine ligand that has low solubility in water.

The transporter adopts an inward open conformation, with a large aqueous channel leading from the cytoplasmic side of the membrane into the central binding site (Fig. 2a). The structure is in a similar conformation to the previous xCT structure[31]. The binding site is flanked on one side by TM1 and TM6, which are broken in the centre of the transporter to from two discontinuous helices, termed 1 A, 1B and 6 A, 6B respectively. At the opposite side of the binding site, TMs 3, 4, 8 & 9 form the hash motif[38], against which TMs 1 and 6 move during alternating access transport[39]. In contrast to the extracellular side, the binding site

| | System xc⁻ (EMD-13267) (PDB 7P9V) | System xc⁻ + glutamate (EMD-13266) (PDB 7P9U) |
|---|---|---|
| **Data collection and processing** | | |
| Magnification | 105,000 | 105,000 |
| Voltage (kV) | 300 | 300 |
| Electron exposure (e−/Å²) | 59.1 | 59.1 |
| Defocus range (μm) | 0.8–2.5 | 0.8–2.5 |
| Pixel size (Å) | 0.832 | 0.832 |
| Symmetry imposed | C1 | C1 |
| Initial particle images (no.) | 8,763,822 | 8,820,414 |
| Final particle images (no.) | 300,221 | 79,698 |
| Map resolution (Å) | 3.4 | 3.7 |
| FSC threshold | 0.143 | 0.143 |
| Map resolution range (Å) | 3.3–4.1 | 3.6–4.7 |
| **Refinement** | | |
| Initial model used (PDB code) | None | None |
| Model resolution (Å) | 3.4 | 3.7 |
| FSC threshold | 0.143 | 0.143 |
| Model resolution range (Å) | 3.3–4.1 | 3.6–4.7 |
| Map sharpening B factor (Å²) | −81.5 | −74.4 |
| **Model composition** | | |
| Non-hydrogen atoms | 7286 | 3933 |
| Protein residues | 919 | 502 |
| Ligands | NAG: 8 | GLU: 1 |
| **B factors (Å²)** | | |
| Protein | 67.17 | 90.04 |
| Ligand | 99.12 | 75.62 |
| **R.m.s. deviations** | | |
| Bond lengths (Å) | 0.005 | 0.003 |
| Bond angles (°) | 0.803 | 0.678 |
| **Validation** | | |
| MolProbity score | 2.28 | 2.02 |
| Clashscore | 15.39 | 11.18 |
| Poor rotamers (%) | 0.00 | 0.00 |
| **Ramachandran plot** | | |
| Favoured (%) | 88.52 | 92.77 |
| Allowed (%) | 11.48 | 6.83 |
| Disallowed (%) | 0.00 | 0.40 |

**Table 1 Cryo-EM data collection, refinement and validation statistics.**

and entrance tunnel are lined with positively charged side chains, Arg48, 135, 340, 396 and Lys198, creating a positively charged environment. This feature is consistent with the requirement to recognise glutamate to drive cystine uptake in this system. An interesting difference with previous SLC7 structures, including the lower resolution xCT structure, is found in TM 6B, which in our structure adopts an unwound conformation (Fig. 2b). However, this feature is clearly visible in the EM maps and exhibits increased stability in the presence of ligand (Supplementary Fig. 5). The unwound formation of TM 6B suggests this part of the intracellular gate undergoes a significant structural change during alternating access transport, consistent with substrate accessibility studies[40]. The extracellular side of the binding site is closed through the close packing of TM1B and 6 A against TM4 and 10, with a strictly conserved aromatic side chain, Tyr244 on

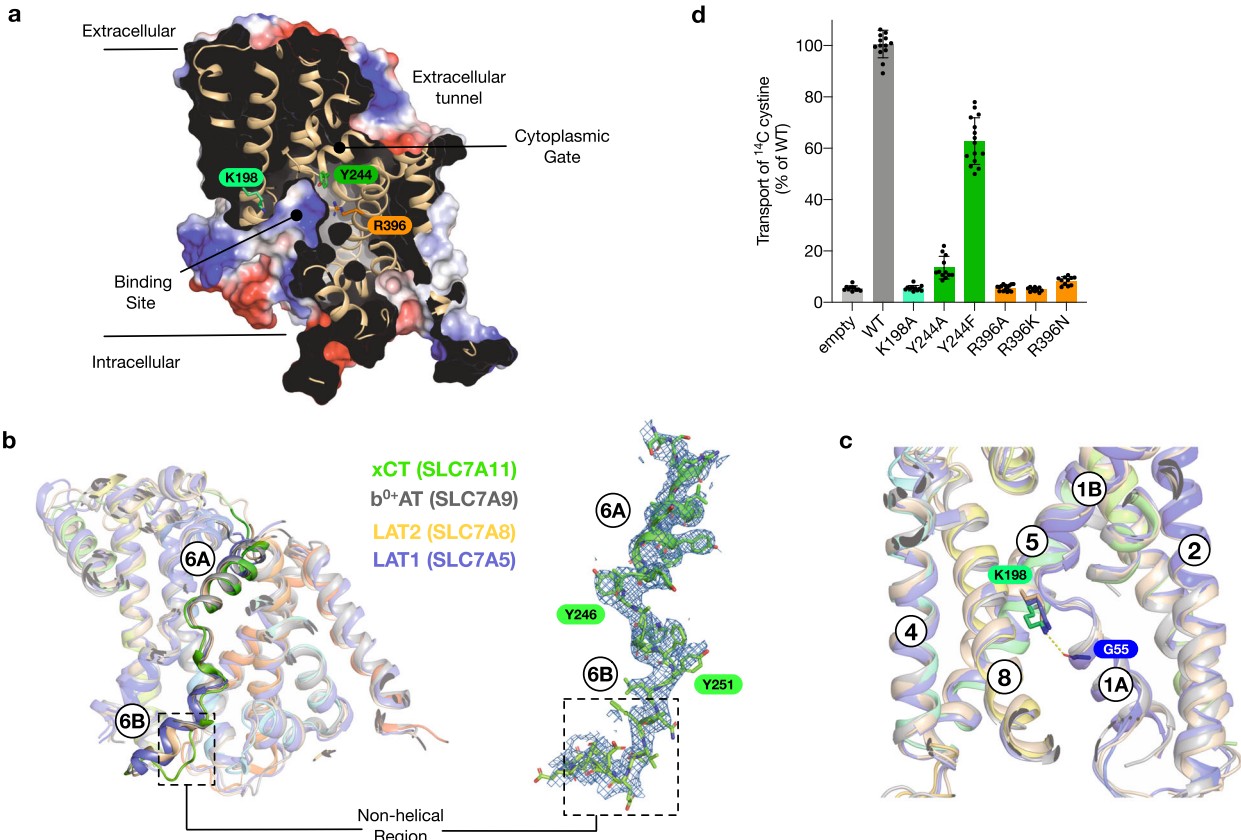

**Fig. 2 Binding site architecture of xCT. a** Electrostatic surface representation highlighting the binding site and key residues. **b** Structural overlay of human SLC7 transporter structures highlighting TM6A and TM6B. The cryo-EM coulombic density for TM6 is shown (blue mesh) for xCT (this study), contoured at 8 sigma. The structural overlay consists of b[0+,]AT (PDB:6lid), xCT (this study), LAT1 (PDB:6irt) and LAT2 (PDB:7cmi). **c** zoomed in view of the interaction between K198 on TM5 and G55 on TM1A in xCT, overlaid are the conserved lysine positions in b[0+,]AT, LAT1 and LAT2, as in **b**. **d** Transport assays of cystine uptake for cells transfected with a plasmid containing wild type or variant forms of SLC7A11 or the empty plasmid as a negative control. $n = 12$ independent experiments, data are presented as mean values and error bars s.d.

TM6A, sealing the binding site entrance. A conserved and essential lysine on TM5, Lys198, which in previous SLC7 family structures acts to coordinate interactions between the unwound region of TM1 and TM8 does not interact with TM8 in xCT[22,34,41]. Interestingly, the equivalent lysine in LAT1, LAT2 and b[0,+]AT also only coordinates TM5 with TM1 (Fig. 2c), suggesting a slightly different role in the eukaryotic members of the SLC7 family. As discussed below, for xCT, the absence of Lys198 interaction would also facilitate the increased structural dynamics observed in TM8 upon ligand binding.

A unique feature of xCT is the presence of Arg396 (Supplementary Fig. 3), which is located on TM10. The presence of this side chain results in a significant increase in the positively charged character of the binding site relative to other SLC7 transporters. In LAT1 and LAT2 this site is occupied by an asparagine, whereas in b[0,+]AT (SLC7A9) it is an alanine. To verify the importance of Arg396 in the transport mechanism of xCT we used a cell-based system to assay cystine uptake (Fig. 2d and Supplementary Fig. 6). Consistent with previously characterised members of the SLC7 family, mutation of Lys198 on TM5 resulted in an inactive transporter, and a large aromatic side chain is required at position 244 on TM6A for the extracellular gate to function[22,34]. Using our assay, we analysed variants of Arg396 to alanine and asparagine, as found in b[0,+]AT, LAT1 and LAT2 respectively (Supplementary Fig. 3). Neither variant displayed transport activity. Even a conservative mutation to lysine was unable to rescue transport, demonstrating that Arg396 is a unique and essential component of the transport mechanism in xCT.

**Structural basis for glutamate recognition.** To understand the molecular basis for substrate selectivity in xCT we determined the structure in complex with L-glutamate (L-Glu) to 3.7 Å (Fig. 3a, Table 1 and Supplementary Fig. 7). Glutamate was observed in the central binding site, with little overall change in the backbone (root mean square deviation 0.70 Å over 448 Cα atoms). Previous structures of both prokaryotic and eukaryotic SLC7 family transporters in complex with ligands have revealed a well conserved mode of binding to amino acids[22,27–29,34,41]. In particular, the amino and carboxy groups of the ligand coordinate with free carboxy and amino groups in the discontinuous regions of TM1 and TM6 respectively, with a water molecule linking the carboxy group to TM7[34]. However, in the structure of glutamate bound xCT we observe the density for glutamate ~ 3.3 Å further away from this canonical binding position, sitting closer to TM3 and 8 from the hash motif (Fig. 3a and Supplementary Fig. 8a, b). The binding of glutamate closer to the hash motif allows for an interaction with Arg135 on TM3 through the gamma carboxylate group (Fig. 3b and Supplementary Fig. 8c). Additional structural changes are observed in the position of Tyr244 on TM6A at the extracellular gate, which moves to hydrogen bond to the carboxylate termini, and Tyr251 on TM6B which interacts with the repositioned Arg135 side chain. Strikingly, the repositioning of Arg135 appears to induce a structural transition of TM8 from an unwound state in the apo structure, to helical in the glutamate bound complex (Fig. 3b, c). In the apo state TM8 adopts a discontinuous structure, with the helix breaking between Gly334 and Ala337. The transition from unwound to helical is also

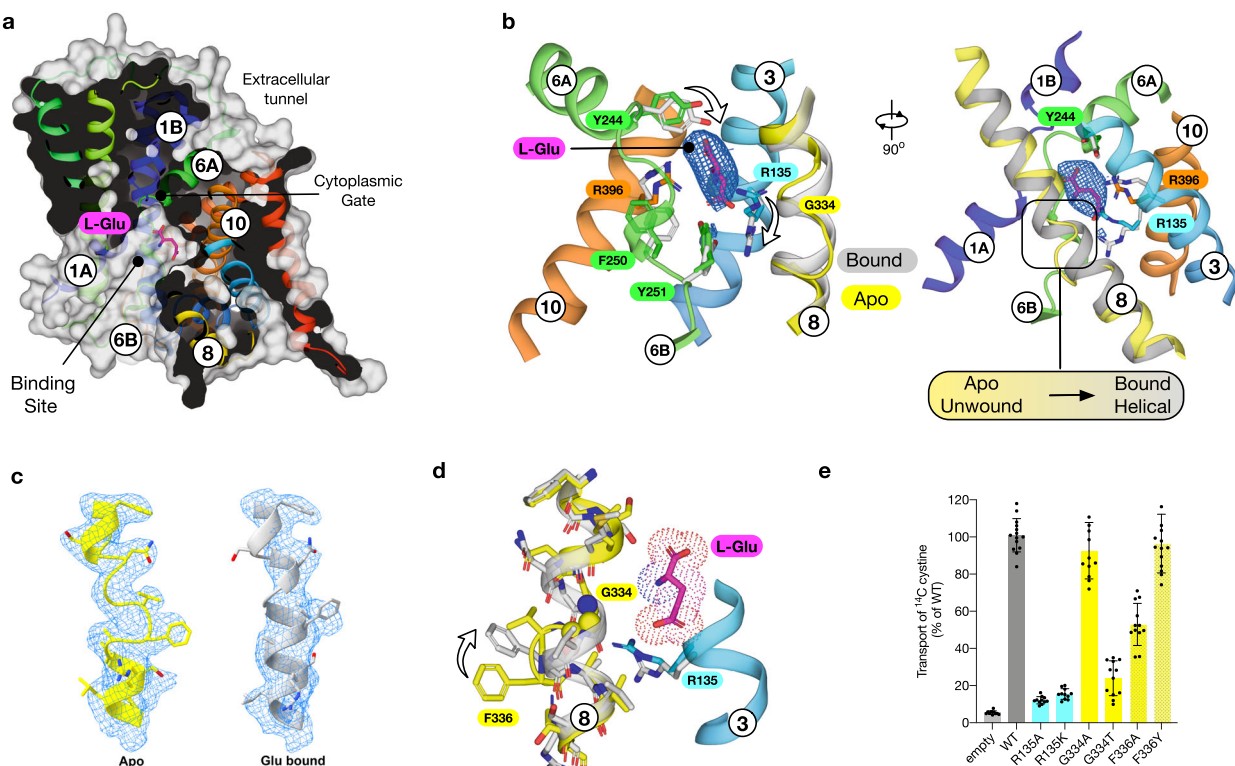

**Fig. 3 L-glutamate bound structure of xCT. a** Molecular surface representation of SLC7A11 showing the bound L-Glu ligand. **b** Zoomed in view of the ligand binding site with differences between TM8 in the apo (coloured) and bound (grey) states. The EM density for the ligand is shown, threshold 8 sigma. **c** Cryo-EM density for TM8 for the apo (yellow) and L-Glu bound (grey) structures. **d** Structural differences on TM8 between bound and unbound forms in relation to the re-positioning of Arg135 on TM3 and subsequent repositioning of Phe336. **e** Transport assays of cystine uptake for cells transfected with a plasmid containing wild type or variant forms of SLC7A11 or the empty plasmid as a negative control. $n = 12$ independent experiments, data are presented as mean values and error bars s.d.

accompanied by the movement of Phe336, which in the apo structure stabilises the discontinuous region through packing against Pro365 on TM7 (Fig. 3d).

To investigate the importance of the residues identified in our structure, we analysed their role in cystine transport (Fig. 3e). Both alanine and lysine variants of Arg135 were inactive, demonstrating the importance of this side chain to the transport mechanism. When interrogating the discontinuity in TM8 we found that only a small side chain was accommodated at position 334, consistent with the observed structural flexibility. However, a conservative substitution of Phe336 to tyrosine displayed WT levels of activity, whereas an alanine variant was ~50 % WT levels, demonstrating an important role for a bulky aromatic side chain in the transition from unwound to helical during ligand binding. Previous structures of SLC7 transporters have reported lipids that sit close to TM8[27,28,34], and these may facilitate the structural rearrangements observed. As discussed below, the structural changes in TM8 observed upon ligand binding suggest an additional mechanism for ligand specificity within this transporter and are consistent with previous observations of an allosteric mechanism in the LAT1 homologue BasC[22].

**Mechanism for cystine specificity.** xCT is unusual within the wider human SLC7 family in being specific for cystine uptake[23]. We therefore wanted to understand how this specificity is achieved for such a chemically distinct amino acid derivative. Using molecular dynamics and guided by previous structures of SLC7 family members, the anionic form of cystine, which predominates at physiological pH[42], was docked into the binding site (Fig. 4a) and the quality of the docking assessed with MD

simulations (Supplementary Fig. 9). Cystine makes several interactions within the binding site that correlate with either previous SLC7 structures or with side chains unique to xCT (Fig. 4b). In common with previous SLC7 members, the α-carboxylate group interacts with the backbone amines of Ala60 and Gly61 in the discontinuous region of TM1. Further interactions are observed to the side chain hydroxyl of Ser330 on TM8 and a water molecule, which coordinates interactions to Tyr281 on TM7 and the discontinuous region of TM1. A similar water molecule is observed in previous SLC7 family structures[22,27–29,34,41] and likely functions in place as the sodium-one ion (named Na+1) found in sodium coupled members of the APC superfamily[43]. Additionally, a second sodium ion (named Na+2) functions to couple TM1 and TM8[44]. In sodium independent members of the superfamily this role is replaced by a conserved lysine on TM5 (Lys198 in xCT)[45]. The interaction with between the α-carboxylate of glutamate and Ser330 on TM8 in xCT may compensate for the loss of the interaction with Lys198 on TM5 and structurally link TM1 and 8. Juxtaposed next to the α-carboxylate, the α-amide group interacts with the backbone carbonyl group of Tyr244, which forms the extracellular gate (Fig. 3b). At the opposite end of the binding site the cystine interacts with Arg135 on TM3 and Arg396 on TM10, which act as hydrogen bond donors and salt bridge partners. The amine group of the cystine sits close to Gly334 on TM8 and makes a hydrogen bond to a water molecule that sits close to Tyr251. Decomposing the frequency of interactions observed over the course of the simulations, the dominant interaction partners to cystine are Arg135 and Arg396 (Supplementary Fig. 9b). Due to the additional electrostatic nature of these interactions, we explored the free energy of cystine binding to quantify the relative

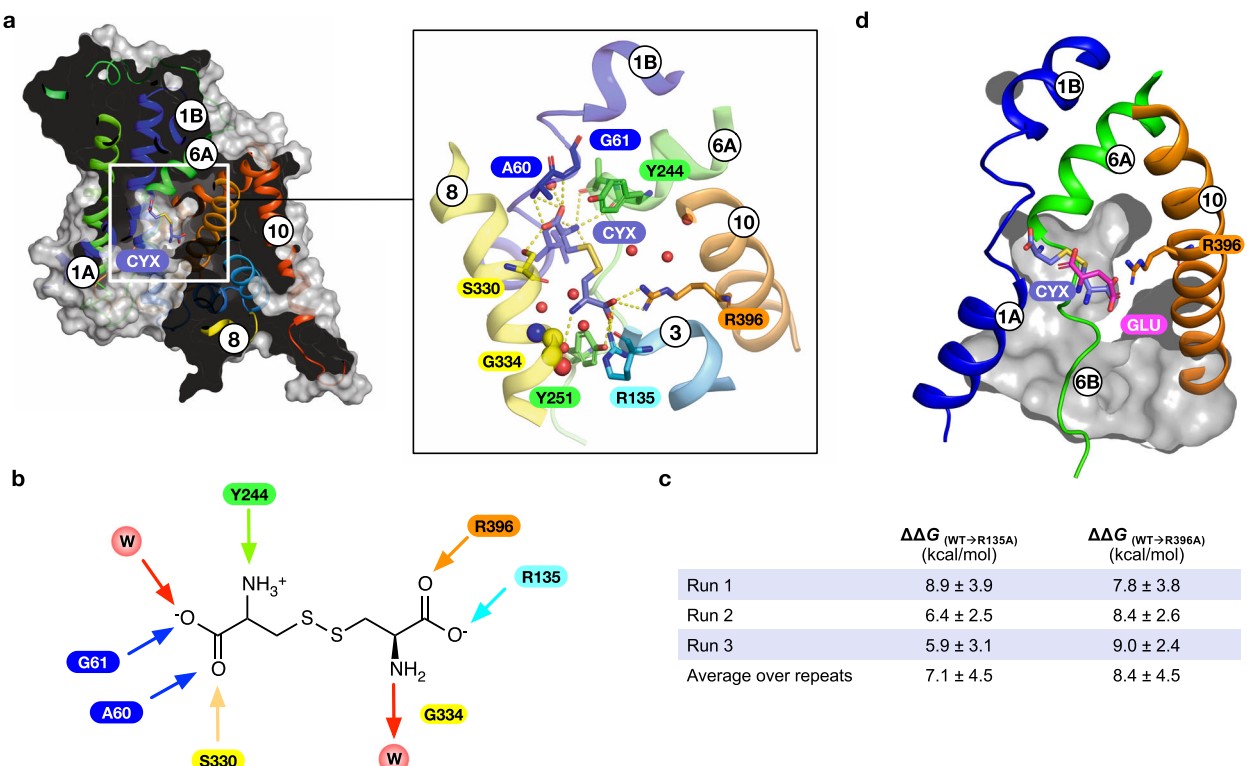

**Fig. 4 Molecular dynamics simulations of L-cystine. a** Slice through the molecular surface representation of xCT indicating the position of final stable pose obtained for cystine (CYX). Inset shows a close-up view of the binding site with polar and hydrogen bond interactions indicated (dashed lines). **b** Schematic of cystine binding interactions, arrows indicate direction of hydrogen bonds. W refers to water molecules, key amino acids are labelled. **c** The calculated binding free energies observed for cystine from three independent simulations (Run1-3) in the WT and alchemically transformed models. **d** Overlay of binding poses for glutamate and cystine.

strength of these interactions (Fig. 4c). Our results suggest that Arg135 and Arg396 contribute equally to the free energy of binding within the error of the method, highlighting the importance of the cationic arginine clamp for cystine specificity. This clamp is unique to xCT and absent from other SLC7 members. The structural changes observed in TM8 following glutamate binding prompted us to analyse the stability of this region of the protein in our cystine-docked structure. Root mean square fluctuations of the backbone residues between 330 and 339 suggest that the binding of cystine reduces the structural flexibility in this region of the helix (Supplementary Fig. 9c), suggesting that cystine binding, similar to glutamate triggers TM8 to adopt a more rigid helical arrangement. This result can be rationalised from the position of cystine in the binding site, as we observe additional interactions to TM8. In particular, the interaction with Ser330 and the close positioning of the β-amine group close to Gly334, which would stabilise the helix in this region. Interestingly we did not observe any direct interactions to the disulphide group in the ligand. However, the electrostatic surface of the binding site is noticeably positively charged, which is contributed by the proximity of the backbone amide of Gly248 in the discontinuous region in TM6 with Arg396 and the indole nitrogen of Trp397.

Cystine contains several unique features that could be used to specifically select for this ligand from the crowded extracellular environment. One feature is the disulphide bond, which contains two sulphur atoms and is negatively charged. The other features are the presence of two carboxylate groups and, at physiological pH, one amide and one amine, both of which are~ 8 Å apart. Previous analyses of inhibitors for xCT have highlighted the importance of different functional groups on the substrate[46]. L-Homocysteate was found to be an effective inhibitor,

demonstrating that the presence of a disulphide is not required for high affinity binding. However, L-Homocysteate is a more effective inhibitor than L-Homocysteine sulphinate, which contains one less oxygen on the terminal sulphoxide group, highlighting the importance of the negative charge at the opposite end of the ligand near Arg135 and 396. The importance of an optimal length of 8 Å for a substrate is demonstrated by the inability of L-Djenkolate, which contains a methylene group between the two sulphur atoms, to effectively inhibit xCT. In comparison, S-Carboxymethyl-L-cysteine, with a length of ~7 Å, is an effective inhibitor. Taken together our findings indicate the ability of xCT to select for cystine is in part driven by the spacing of two opposing negative charges ~7–8 Å apart, which can coordinate the mobile gate helices (TM1 & 6) with the hash motif helices (TM3 & 10).

Finally, when comparing the positions of the two ligands in our structures we observe that γ-carboxylate group of L-glutamate sits in the same position as the second carboxylate in cystine, sitting close to Arg135 and Arg369 (Fig. 4d). The key difference in the binding positions is that glutamate does not engage with the canonical binding site in TM1 and TM2, discussed above. We suggest that the reason for this is due to the nature of the allosteric effect observed in TM8 (Fig. 3b), which as discussed below potentially has important implications for the transport mechanism in this system.

## Discussion

An important finding from this work is the observation that structural dynamics in TM8 plays an important role in the transport mechanism in xCT. Our MD simulations with bound L-cystine also reveal that TM8 remains helical in the region directly

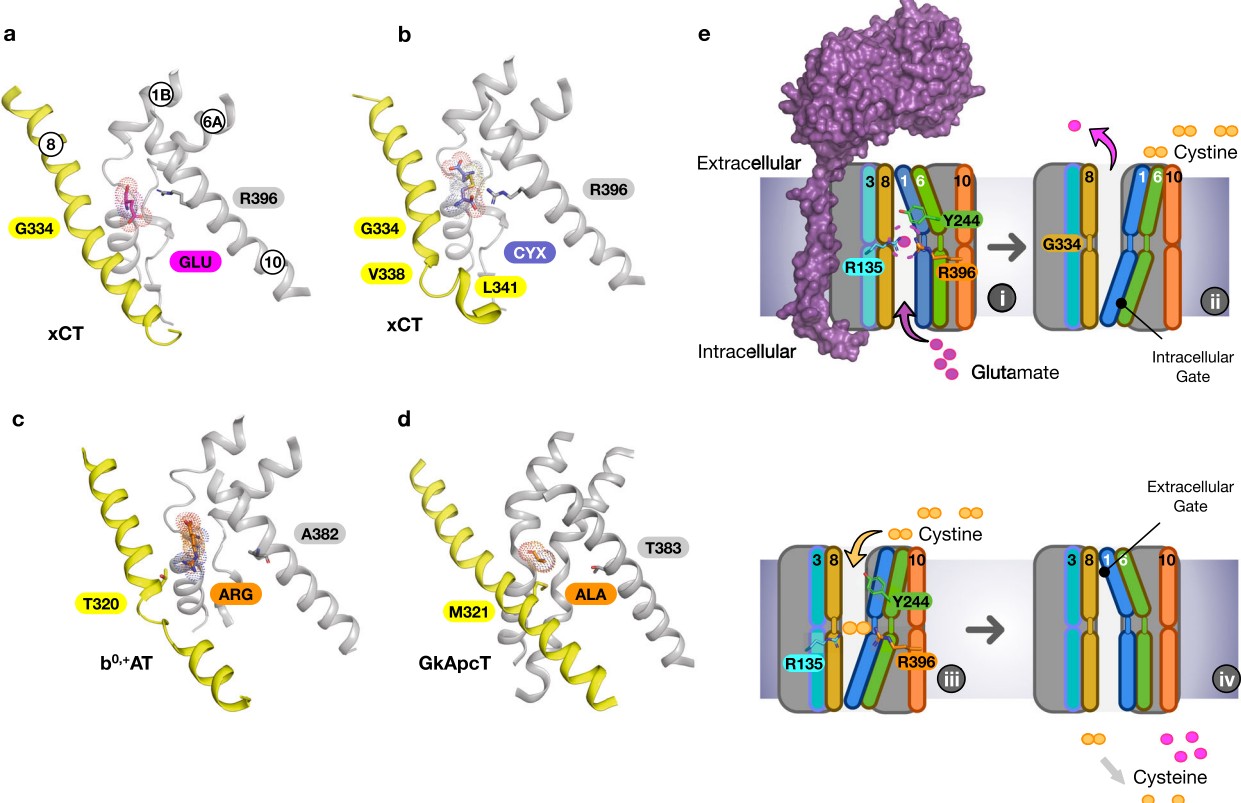

**Fig. 5 Mechanism for cystine-glutamate recognition and antiport by system xc⁻.** Comparison of the binding site in xCT bound to L-glutamate (**a**) and L-cystine (**b**) with the L-Arginine bound b0+,AT (PDB:6li9) (**c**) and GkApcT (PDB:5oqt) (**d**) highlighting the structural differences in TM8. Side chains discussed in the text are highlighted. **e** Key steps in the transport cycle for system xc⁻, as discussed in the text.

opposite the ligand (Gly334) but loops out towards the cytoplasmic end (Val338–Leu341), stabilised through a salt bridge between Arg340 and Glu497 at the C-terminus. This raises the question of the role of TM8 in substrate selectivity. Comparison of the SLC7 family structures, including prokaryotic homologues, reveals that TM8 remains helical in almost all structures (Fig. 5a–d). The exception is b0+,AT, which also displays an unwound segment of TM8 and similar to xCT also transports L-cystine, in addition to other cationic and neutral amino acids[47]. The effect of the unwound region in b0+,AT is to create a larger binding pocket relative to LAT1 and LAT2, which would facilitate the binding of longer ligands, such as lysine, arginine and cystine. The presence of a threonine side chain at the equivalent position to Gly334 in xCT may also contribute to its broader substrate profile, including cationic and neutral amino acids[48], by introducing additional rigidity and hydrogen bond interactions in this region of the binding site. In xCT it is likely a similar mechanism is at play, except that TM8 is functioning to dynamically modulate the binding site, rather than function in a static manner observed in previous SLC7 family members. In the presence of L-glutamate TM8 adopts a conformation that induces structural rigidity to facilitate the positioning of the small ligand. Whereas to accommodate L-cystine, the binding site needs to expand, which is achieved through the unwinding of the cytoplasmic half, similar to b0+,AT.

The mechanistic trick that xCT performs is to select only cystine from the extracellular amino acid pool. The docking results indicate that Arg396 on TM10 is a key determinant of specificity, as it functions to coordinate the second carboxylate and push the ligand into the canonical binding pose with TM1 and 6. Our functional assays confirm the necessity of arginine in this position (Fig. 2d), demonstrating that additional structural

contributions, possibly involving the stabilisation of TM8 following ligand binding, are required in addition to its cationic properties. An equally important role is played by Gly334, which mediates the flexibility of TM8 and enables accommodation of the neutral amine group of L-cystine. Interestingly, the equivalent position to Gly334 in related APC family transporters also impacts substrate binding[41,49]. For example, in GkApcT, Met321, which sits in the same position as Gly334 in xCT (Fig. 5d), functions to control substrate selectivity through steric constraints[34], similar to the mechanism we propose here for TM8. The dual mechanism for ligand specificity, combining the electrostatic clamp between the two carboxylate groups in the ligands and steric complementarity following binding, ensures that xCT can only recognise either L-cystine or L-glutamate. The relative concentrations of these amino acids in the body dictates the transport direction. Finally, the observation that L-glutamate functions to link TM1 and TM8 is consistent with previous observations in the sodium-independent arginine/agmatine exchanger AdiC, where the substrate guanidinium group interacts with a tryptophan on TM8 to increase substrate affinity and facilitate transport[41,49].

Taken together we can propose a working model for cystine/glutamate transport via system xc⁻ (Fig. 5e). Starting from our structure of the glutamate bound inward open state (i) we propose that ligand binding causes the allosteric transition in TM8, which facilitates glutamate recognition and closure of the intracellular gate, via the packing of TM1A and 6B against TM8. In the outward open state, the transporter will release glutamate (ii), whereupon we speculate that TM8 may adopt an unwound state around Gly334, similar to that observed in our inward open apo structure (Fig. 3c) and facilitating L-cystine entry (iii). Cystine functions to coordinate the mobile gate helices, TM1 and 6 with

the hash domain. Specifically, Arg135 on TM3 and Arg396 on TM10, which pull Tyr244 on TM6A to seal the cytoplasmic gate. TM1A then moves away from the hash domain to open the binding site to the cytoplasm, allowing cystine to exit (iv), whereupon it is immediately reduced to L-cysteine. Given the high concentrations (mM) of L-glutamate in the cytoplasm, L-glutamate will rapidly enter the binding site, completing the antiport cycle (i).

Finally, inhibiting xCT is seen as a promising route to tackling and managing several different cancers, including pancreatic, gastrointestinal, colorectal, and glioblastoma[16,50]. However, to date specific inhibitors that target xCT without significant cross reactivity with other metabolic pathways, remain to be discovered[51,52]. The structural insights into cystine recognition provided by this study, combined with the first substrate model for xCT (Fig. 4b and Supplementary Fig. 8c), should enable a more rational approach to inhibiting system xc−.

## Methods
### Cloning, expression and purification of SLC7A11/SLC3A2.
The gene encoding human SLC7A11(I.M.A.G.E. clone IRAUp969G0966D) was cloned into pLexM (Addgene 99844)[53] with an N-terminal Flag tag inserted via PCR. SLC3A2 (I.M.A.G.E. clone IRAUp96D0818D) was also cloned into pLexM with an N-terminal His$_8$ tag inserted via PCR, the sequence of the oligonucleotides used can be found in the supplementary material as Supplementary Table 1. The two plasmids were transiently co-transfected into HEK293F cells in FreeStyle™ 293 Expression Medium (ThermoFisher Scientific, UK) with a low passage number (less than 10), as higher passage number resulted in poor quality protein. The cells were cultured in suspension phase at 37 °C, 8% CO$_2$. 18–24 h before transfection, cells were passed at a density of $7 \times 10^5$ cells/ml to give a density of 1.3–1.4 $\times 10^6$ cells/ml at transfection. For 2 L culture, 2 mg of plasmid DNA (1 mg HsSLC7A11 + 1 mg HsSLC3A2) was diluted in Dulbecco's Modified Eagle Medium (DMEM, ThermoFisher Scientific, UK) to a total volume of 30 ml, it was mixed gently and left to stand for 5 min. In that time, 4 ml (from 1 mg/ml stock) of linear polyethyleneimine (PEI) MAX (Mw 40,000; Polysciences Inc.) was diluted in the same medium to a volume of 30 ml in a separate tube and mixed. The diluted PEI MAX was added to the diluted DNA and mixed gently. The mixture was incubated for 10–15 min at room temperature and was added drop-wise to the cells with gentle swirling. Sodium butyrate was then added at 8 mM final concentration. Cells were returned to the incubator and harvested 36 h post-transfection, expression for longer time periods (48 or 72 h) resulted in lower quality protein.

The cell pellet was thawed and resuspended in ice-cold PBS containing DNase and cells were subsequently lysed using a Sonicator (Q Sonica, USA), 8 cycles of 30 s on followed by 60 s off on ice. Unbroken cells and cell debris were pelleted at 10,000 g for 10 min at 4 °C and membranes were harvested through centrifugation at 200,000 g for one hour and washed once with 20 mM HEPES pH 7.5, 20 mM KCl. After washing the membranes were resuspended in PBS and snap frozen for storage at -80 until required. Membranes were thawed and solubilized in 1× PBS, 150 mM NaCl, 10% glycerol containing 1% LMNG:CHS (5:1 ratio) for 90 min at 4 °C. Insoluble material was removed through centrifugation for one hour at 200,000 g. SLC7A11/SLC3A2 was purified to homogeneity using standard immobilized metal-affinity chromatography protocols in LMNG:CHS followed by flag affinity chromatography. The final protein complex was subjected to size exclusion chromatography (Superose 6 10/300) 20 mM Tris pH 7.5, 150 mM NaCl with 0.01 % LMNG and 0.002 % CHS. Other detergents were trialled, DDM:CHS, GDN and LMNG with no CHS, while protein could be purified in both GDN and DDM:CHS the yield using DDM:CHS was very low and the grids made from the GDN sample failed to produce a structure.

### Cell based transport assays.
HeLa cells were maintained in RPMI 1640 medium supplemented with 10 % foetal bovine serum and 2 mM L-glutamine under 5% CO$_2$ at 37 °C. For transport assays 2 $\times 10^5$ cells per well were seeded into 12 well plates and 24 h later transfected using lipofectamine 2000 with equal amounts of SLC7A11/SLC3 plasmids for 36 h. Cells were washed twice with 1 ml of PBS before application of 0.4 ml PBS containing cystine with trace amounts of $^{14}$C cystine (2 µM). After the desired time the assay buffer was removed, and the cells quickly washed twice with 0.5 ml assay buffer with no cystine. Cells were removed using trypsin (0.1% in PBS for 2 min) and placed in a scintillation vial containing 100 ul 1 M NaOH and lysed for 5 min prior to the addition of scintillation fluid. The amount of cystine taken up by the cells was calculated by scintillation counting in Ultima Gold (Perkin Elmer) with comparison to a standard curve. Experiments were performed a minimal of eight times to generate an overall mean and s.d. Assay data was analysed and graphically represented using Prism version 9.

### Cryo-EM sample preparation and data acquisition.
System xc− (5 mg/ml in LMNG:CHS) incubated in the presence or absence of 6.8 mM glutamate was adsorbed to glow-discharged holey carbon-coated grids (Quantifoil 300 mesh, Au R1.2/1.3) for 10 s. Grids were then blotted for 2 s at 100% humidity at 8 °C and frozen in liquid ethane using a Vitrobot Mark IV (Thermo Fisher Scientific). Data were collected in counted super-resolution mode on a Titan Krios G3 (FEI) operating at 300 kV with a BioQuantum imaging filter (Gatan) and K3 direct detection camera (Gatan) at 105,000× magnification, physical pixel size of 0.832 Å. 15,504 and 14,519 movies were collected for system xc− with or without glutamate, respectively. Movies were collected at a dose rate of 22.2 e-/Å$^2$/s, exposure time of 2.66 s, corresponding to a total dose of 59.1 e-/Å$^2$ split over 40 fractions.

### Cryo-EM data processing.
Initial micrograph processing was performed in real time using the SIMPLE pipeline[54], using SIMPLE-unblur for patched (15 × 10) motion correction, SIMPLE-CTFFIND for patched CTF estimation and SIMPLE-picker for particle picking. After initial 2D classification in SIMPLE to remove junk particles, all subsequent processing was performed in either cryoSPARC[55] or RELION-3.1[56] using the csparc2star.py script within UCSF pyem[57] to convert between formats. Resolution estimates were derived from gold-standard Fourier shell correlations (FSCs) using the 0.143 criterion as calculated within RELION, cryoSPARC or the remote 3DFSC[58] processing server. Local resolution estimations were calculated within RELION.

For system xc− (Supplementary Fig. 2), a subset of 866,446 particles following initial 2D classification in SIMPLE were subjected to multi-class 3D initial model generation within RELION (k = 5). The resulting volumes, lowpass-filtered to 25 Å, were used as references for supervised 3D classification (20 iterations at 7.5° sampling followed by 10 iterations at 3.75° sampling) against the same particle subset. The resulting maps were then lowpass-filtered to 15 Å and used as references for supervised 3D classification (20 iterations at 7.5° sampling) against the full 2D-classified particle set (6,401,560 particles). This produced a single class with clear transmembrane helices. Particles (2,404,959) belonging to this class were further 3D-classified (k = 4, 20 iterations at 7.5° sampling) against its corresponding volume (lowpass-filtered to 15 Å) with a mask encompassing the full particle (protein plus detergent). The predominant particle subset (69.0%) was then subjected to 3D auto-refinement against its own map (lowpass-filtered to 15 Å) to yield a 3.9 Å reconstruction. These particles were Bayesian polished and further classified in 2D (k = 200) to generate a subset of 903,265 cleaned and polished particles. These particles were then imported into cryoSPARC and subjected to a multi-class ab initio reconstruction (k = 4) to improve particle homogeneity. Selected particles (300,221) from the strongest class were used for non-uniform refinement in cryoSPARC against the non-polished 3.9 Å map (lowpass-filtered to 8 Å), improving the reconstruction to 3.6 Å. This density was further improved to 3.4 Å by local non-uniform refinement in cryoSPARC using prior orientations, an 8 Å low-pass-filtered reference and a soft mask encompassing only protein.

For system xc− in complex with glutamate (Supplementary Fig. 7), 6,115,752 particles were retained after initial 2D classification in SIMPLE. These particles were subjected to supervised 3D classification (20 iterations at 7.5° sampling) against the same five initial volumes generated from the apo system xc− dataset, lowpass-filtered to 15 Å. The class with strongest transmembrane helix density, containing 2,395,948 particles, was further subjected to 3D classification (k = 4, 20 iterations at 7.5° sampling) against its corresponding volume lowpass-filtered to 15 Å. Particles (1,557,854) selected from the dominant class were subjected to 3D auto-refinement against its own map (lowpass-filtered to 15 Å), resulting in a 4.1 Å reconstruction. These particles were Bayesian polished and further classified in 2D (k = 200), generating a subset of 964,303 cleaned and polished particles. These particles were then imported into cryoSPARC and further pruned by heterorefinement against four 8 Å lowpass-filtered volumes generated from the cryoSPARC multi-class ab initio reconstruction of apo system xc−. Selected particles (380,184) from the strongest class were subjected to non-uniform refinement in cryoSPARC against their corresponding volume (lowpass-filtered to 8 Å), yielding a 3.5 Å map. Local non-uniform refinement, using prior orientations, was performed against an 8 Å low-pass-filtered reference and soft mask encompassing only the transmembrane domain of system xc−, generating a 3.0 Å volume for this region. Particle stacks and orientations were then re-imported into RELION and subjected to alignment-free classification (k = 6, T = 20) using a soft mask that encompassed the putative ligand binding site. Particles (79,698) belonging to a single class with unambiguous ligand density were imported back into cryoSPARC and used for global non-uniform refinement using the 3.5 Å polished map as reference (lowpass-filtered to 8 Å). This generated a 3.9 Å volume with clear ligand density; local non-uniform refinement using masks encompassing the transmembrane domain or full protein generated focused reconstructions at 3.7 Å and 3.6 Å, respectively.

### Model building and refinement.
Apart from the ectodomain of heavy chain 4F2hc, which was guided by PDB 6IRS, the atomic model of system xc− (Table 1) was built de novo from the globally-sharpened 3.4 Å map following multiple rounds of manual building using Coot v. 0.9[59] and real-space refinement in PHENIX v. 1.18.2-3874[60] using secondary structure, rotamer and Ramachandran restraints.

The atomic model of system xc⁻ in complex with glutamate was generated by rigid body fitting system xc⁻ into the globally-sharpened focused reconstruction of the transmembrane domain (3.7 Å) followed by multiple rounds of real-space refinement in Coot and PHENIX. Glutamate ligand was imported from Coot monomer libraries and rigid body fit into map density, followed by further rounds of real-space refinement in Coot and PHENIX. Both system xc⁻ models (apo and glu-bound) were validated using MolProbity[61] within PHENIX. Figures were prepared using UCSF ChimeraX v.1.1[62], PyMOL v.2.4.0 (The PyMOL Molecular Graphics System, v.2.0; Schrödinger) and BioRender.com (2020).

**Docking and molecular dynamics.** The full heterodimer of system xc⁻ (xCT-4Fhc2) was embedded into a pure 1-palmitoyl-2-oleoyl-sn-glycero-3-phosphocholine (POPC) membrane via CHARMM-GUI[63]. The heterodimer contained 500 lipids on both leaflets, with 258,000 atoms in a $12.5 \times 12.5 \times 16.0$ nm³ simulation box. In the cystine-bound simulations, the transporting SLC7A11 domain was simulated with 500 lipids, totalling 227,000 atoms in a $13.5 \times 13.5 \times 12.2$ nm³ simulation box.

The coordinates of b⁰,⁺AT-rBAT+Arg complex [PDB: 6li9] were obtained, the rBAT domain was deleted and the transporting b⁰,⁺AT domain superimposed to the apo xCT-4Fhc2 cluster average structure obtained from 200 ns simulation. Hydrogen atoms were added to cystine according so that cystine had a -1 net charge, according to a previous publication and similar to the glutamate substrate[42]. The SLC7A11-Arg complex was saved and was subsequently used as a template to guide docking of the cystine ligand to SLC7A11. We used pre-existing structural data of amino acid binding interactions to other SLC7 members to guide our docking and increase confidence. Specifically, the (-COO⁻, -NH₃⁺ groups) belonging to half of cystine were restrained to 2 Å of the arginine -COO⁻, -NH₃⁺ group coordinates. Additionally, H-bonds with I57, G61, Y244 and the crystallographic water molecule from the GkApcT structure were set as docking constraints. The docking grid was generated with Glide assigning the OPLS3e force field to protein and cystine[64]. Only the anionic cystine was considered as this is the only form of cystine present in physiological conditions[42]. The resulting poses were degenerate and converged within 1 Å. The top-ranked pose according to the scoring function Glide XP[64] was saved and used as starting structure for MD simulations.

For MD simulation the protein was represented with the Amber99SB-ILDN forcefield[65] and the lipid was represented with the Slipid forcefield[66]. Cystine was represented with the General Amber Force Field force field[67] and the charge was derived using the semiempirical AM1-BCC method bundled with AmberTools18. The system was then solvated with TIP3P water[68] and neutralised to an ionic concentration of 0.15 M with NaCl.

MD simulations in explicit solvent used Gromacs 2020.3[69] as the MD engine. The electrostatic interactions were computed using particle mesh Ewald[70] with a short-range cut-off of 1.2 nm, which was also used for the Lennard–Jones interactions cut-off. Bonds involving hydrogen atoms were constrained by the LINCS algorithm[71], and a time step of 2 fs was used for the integration of the equations of motion. Snapshots recorded every 10 ps during the production MD simulations were considered for analysis. The system was minimised with steepest descents and a conjugate-gradient step every 10 steps until convergence to $F_{max} < 500$ kJ/mol. After energy minimisation, the system was equilibrated with 1000 kJ mol⁻¹ nm⁻² positional restraints on the heavy atoms for 5 ns with the v-rescale thermostat[72] at 310 K in the NVT ensemble. The system was further equilibrated with the same positional restraint in the NPT ensemble with a semi-isotropic Berendsen barostat[73] at 1 atm and a Nose–Hoover thermostat at 310 K for 10 ns. The extended equilibration in the NPT ensemble ensured the potential energy of the system and box volume converged. The resulting frame was used as the starting frame for the production runs. Initial velocities were sampled from the Maxwell-Boltzmann distribution for each repeat. Production runs were performed for 200 ns with the Parrinello–Rahman barostat[74] and the Nose–Hoover thermostat at 310 K.

**End-point molecular mechanics-Poisson Boltzmann surface area calculations (MMPBSA-TΔS).** MM-PBSA is an intuitive approach that combines molecular mechanics and continuum models to calculate end-point binding free energies. The polar part of the solvation free energy was calculated using Poisson-Boltzmann (PB) equations. In these calculations, a dielectric constant of $\varepsilon_{solute} = 4$ was assigned to the binding area owing to the highly charged residues present, and $\varepsilon_{solute} = 80$ for water. A variable dielectric was assigned along the membrane normal, as demonstrated here[75]. The binding free energy for each complex was calculated using Eq. (1)

$$\Delta G_{eff} = \Delta E_{MM} + \Delta G_{sol} - T\Delta S_{MM} \qquad (1)$$

In Eq. (1) $\Delta G_{eff}$ is the binding free energy for each calculated complex. The entropic contribution, $\Delta S_{MM}$, to binding free energy was calculated with the interaction-entropy method, which has sound roots in statistical mechanics, shown here[76]. Explicit entropic contributions to protein-ligand binding free energies have been shown to significantly improve correlation with experimental measurements[77]. $\Delta E_{MM}$ defines the interaction energy between the protein and the ligand as calculated by molecular mechanics in the gas phase. $\Delta G_{sol}$ is the deso-lvation free energy for transferring the ligand from water to the binding area

calculated, using the PBSA model. The terms for each complex $\Delta E_{MM}$ and $\Delta G_{sol}$ are calculated using Eqs. (2) and (3)

$$\Delta E_{MM} = \Delta E_{elec} + \Delta E_{vdW} \qquad (2)$$

$$\Delta G_{sol} = \Delta G_P + \Delta G_{NP} \qquad (3)$$

In Eq. (2) $\Delta E$elec and $\Delta E$vdW are the electrostatic and the van der Waals inter-action energies, respectively. In Eq. (3) $\Delta G_P$ is the electrostatic or polar con-tribution to the free energy of solvation and the term $\Delta G_{NP}$ is the non-polar or hydrophobic contribution to the solvation free energy. For MM-PBSA calcula-tions, molecular mechanics energies and the non-polar contribution to the sol-vation free energy were computed with the *mmpbsa.py* module of AmberTools20[78]. *gmx-MMPBSA* (v1.4.3)[79] was used to convert the Gromacs trajectories into AMBER format with *parmed*, and subsequently used *mmpbsa.py*. Alanine scanning mutations of R135 and R396 were done to quantitatively characterise important residue contributions, as shown previously with MMPBSA[80]. Data post-processing of binding free energies from MMPBSA was done with the *gmx_MMPBSA_ana* analysis module. Even though MMPBSA is known to be less accurate than some of the more computationally expensive methods, such as the free energy perturbation and thermodynamic integration, the qualitative agreement is accurate for the ranking of protein-ligand interactions (by including solvation and entropic terms, as opposed to a simple molecular mechanics energy calculation). Importantly alanine scanning MMPBSA has been used with good success in identifying key residues for binding[81] justifying the selection of MMPBSA to decompose the relative contributions of Arg135 and Arg396 to the binding free energy.

MMPBSA approximately calculates the binding free energy of host-guest systems. In Fig. 4c the relative $\Delta\Delta G$ values are shown which required careful statistical processing of the binding free energy values. Regarding statistics shown in Fig. 4c, to calculate mean of means the usual formula was used:

$$\bar{x} = \sum_{k=1}^{m} \frac{n_k}{n} \cdot \bar{x}_k, \ where \ n = \sum_{i=1}^{m} n_i \qquad (4)$$

and $n_1 = ... = n_2$, the sample sizes from production runs of either independent repeat of WT/mutant systems were the same in all cases.

For the calculation of the pooled standard deviation of the difference of the means of binding free energies between WT and mutant, $1 \times M$ vectors consisting of energy values from selected frames, $M$, were subtracted and the pooled standard deviation was calculated from the components of the resultant difference vector. The formula used for calculations of the standard deviation of pooled means (mean of the means of each run) was:

$$\hat{\sigma}_{\bar{x}} = \sqrt{\frac{(N_1-1)SD_1{}^2 + (N_2-1)SD_2{}^2 + ... + (N_k-1)SD_k{}^2}{(N_1-1) + (N_2-1) + ... + (N_k-1)}}, \qquad (5)$$

where $N_k$ indexes the statistical degrees of freedom of each sample (i.e., the number of analysed frames), and with the assumption that the independent MD runs are statistically independent. This assumption is justified due to seeding production simulations replicates with velocities randomly selected from the Maxwell-Boltzmann distribution. Therefore, the covariance of the samples is taken as 0 in all cases, and Eq. (5) holds. Since the number of samples (i.e., MD frames selected for MMPBSA calculations) were the same in all cases, $N_{run1} = N_{run2} = N_{run3}$, the Eq. (5) simplifies to.

$$\hat{\sigma}_{\bar{x}} = \sqrt{\frac{SD_1{}^2 + SD_2{}^2 + SD_3{}^2}{3}} \qquad (6)$$

**Reporting summary**. Further information on research design is available in the Nature Research Reporting Summary linked to this article.

## Data availability
Atomic coordinates and maps for system xc⁻ have been deposited in the PDB and Electron Microscopy Data Bank under IDs 7P9V and EMD-13267 for the apo state and 7P9U and EMD-13266 for the glutamate bound state. Source data are provided with this paper. All other relevant data are available from the corresponding authors upon reasonable request. Source data are provided with this paper.

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

## Acknowledgements

The Central Oxford Structural Microscopy and Imaging Centre is support by the Wellcome Trust (201536) to S.M.L. and S.N. The EPA Cephalosporin Trust, and a Royal Society/Wolfson Foundation Laboratory Refurbishment Grant (WL160052) to S.M.L. and S.N. Computing was supported via the Advanced Research Computing facility, Oxford, the ARCHER UK National Supercomputing Service and JADE (EP/T022205/1) granted via the High-End Computing Consortium for Biomolecular Simulation, (HECBioSim - http://www.hecbiosim.ac.uk), supported by EPSRC (EP/R029407/1) to P.C.B. This research was supported by Wellcome awards to SML (209194;100298), P.C.B. (219531) and S.N. (215519;219531) and through MRC grants to S.M.L. (MR/M011984/1) and J.L.P. (MR/S021043/1). D.K. is a UKRI BBSRC DTP PhD student.

## Author contributions

J.L.P. and S.N. conceived the project. G.K. maintained cell stocks, undertook large scale expression and tissue culture. J.L.P performed all protein preparation. J.C.D. and S.M.L. performed all cryo-EM sample processing, data collection and image analysis. J.C.D., S.M.L. and S.N. constructed the atomic models. J.L.P. conducted all transport assays. D.K. and P.C.B. performed all molecular dynamics analysis. J.L.P. and S.N. wrote the manuscript and prepared figures with contributions from all authors.

## Competing interests

The authors declare no competing interests.
