## [Peer Review File · Nature Communications]

Molecular basis for redox control by the human cystine/ glutamate antiporter System xc-REVIEWER COMMENTS

Reviewer #1 (Remarks to the Author):

The manuscript by Parker et al. presents cryo-EM structures of apo and glutamate-bound states of human xc- transporter, accompanied by functional analysis of WT and mutants, as well as in silico analysis of the cysteine-bound state. Despite several structural reports on SLC7 transport systems, this manuscript uncovers important details of the xc- transport mechanism, relevant in human health and disease, notably the first substrate-bound xc- structure, and a major improvement in resolution compared to a previous report. As such, I found this study of great interest for the membrane transport field, and recommend publication.

The manuscript is clearly written, and the experimental design, as well as structural/functional analyses are sound. The comments below aim at improving clarity, or correcting mistakes:

1-Regarding sample preparation: in the first “results” section, it is mentioned that protein expression/purification, as well cryoEM sample preparation went through several rounds of optimisation. The “methods” section describes protocols in detail but there is no mention of optimisation strategies. It would be useful to know what was optimised, and what seemed to be important, as previous attempts to purify an xc-transporter used a different strategy based on consensus mutagenesis, and led to a lower resolution cryoEM structure. Also, since xc- is a cystine/glutamate exchanger one would have expected glutamate- and cystine-bound states, respectively, as targets for structural studies. A comment on why the cystine-bound complex was not pursued, or failed to yield high-resolution structures would again be informative.

2-Regarding ectodomain/SLC7A11 interface, there is a mistake in Fig.1d, where instead of Q217 and Q219, it reads R135 twice. More importantly, at the end of page 5 the authors mentioned that a 20-degree opening compared to LAT1 “results in a wider vestibule, no doubt to accommodate the larger cystine ligand. SLC7A11 also has a noticeably less charged tunnel leading to the extracellular gate compared to LAT1 and LAT2, which may also facilitate binding of the hydrophobic cystine ligand.” I think these statements should be explained in more depth. The vestibule in published LAT1 structures is wide enough to let cystine enter, although it is not a substrate, so why do the authors think that further widening of the vestibule is specifically associated to cystine binding? How does the vestibule seen in their structure compare to the published ~6 angstrom xc- structure? Could it be that the ectodomain is dynamic relative to the transporter? I would expect so, since they are linked through a disulfide bond, and likely weak contacts. Have the authors explored cryoEM 3D-variability or multi-body analyses to look into conformational heterogeneity?

Finally, in the section dealing with cystine binding, the authors mention that physiologically relevant cystine bears a formal negative charge, plus several polar alpha-amino and -carboxylate groups. Why is it described here as “hydrophobic cystine ligand”?

3-I found the graphical part of the manuscript a bit frugal, and some efforts could be put towards improving this part. For example: in figures throughout the main text the limits of the membrane are not included, and transporter orientations (Figs. 2-4) leave the impression that the bottom of the extracellular vestibule reaches halfway across the hydrophobic core of the membrane. Cryo-EM maps contain clear density for the micelle and the authors could use them to estimate membrane “borders”. Also, one of the main differences observed in xc- apo structure compared to previous SLC7 structures is the extended conformation of TM6b, but this is shown in a supplementary Fig (Supp Fig 5). The same applies to K198 position. Why not to highlight these structural aspects in main-text figures?

4-There is a typo at the end of page 7 “Consistent with previously members of the SLC7 family,”.

Reviewer #2 (Remarks to the Author):

Molecular basis for redox control by the human cystine/glutamate antiporter System xc

Parker et al.

Parker et al present a structural study of the human xCT cystine/glutamate antiporter which suggests a mechanism for ligand recognition and transport. There are a few issues that I would like addressed before publication.

I am curious to learn why the authors have not attempted to capture the antiporter in the presence of cystine, instead of relying solely on molecular dynamics.

There appear to be several novel secondary structure transitions that occur in xCT during transport. The authors mention the conformation of TM6B, but this is not explored further so it is never confirmed if this feature is relevant in protein function. Given that this appears to be one of the unique structural features of the human xCT, I think the manuscript would be strengthened by a closer look at the importance of this segment in its function.

In general, I would like to see the main figures relate better to the manuscript text. Specifically, I would like the authors to make the following changes:

Figure 1

- Include a panel to show a better representation of the protein topology. Despite this being a common fold, the readers would benefit from a good illustration.
- Panel 1 d- the residues are mislabeled.

Figure 2

- I would strongly recommend adding panels to illustrate the structure described in the text: "The binding site is flanked on one side by TM1 and TM6, which are broken in the centre of the transporter to form two discontinuous helices, termed 1A, 1B and 6A, 6B respectively. At the opposite side of the binding site, TMs 3, 4, 8 & 9 form the hash motif 38, against which TMs 1 and 6 move during alternating access transport."

And

"The extracellular side of the binding site is closed through the close packing of TM1B and 6A against TM4 and 10, with a strictly conserved aromatic side chain, Tyr244 on TM6A, sealing the binding site entrance. A conserved and essential lysine on TM5, Lys198, which in previous SLC7 family structures acts to coordinate interactions between the unwound region of TM1 and TM8 does not interact with TM8 in xCT"

Figure 3

In panel 3c the glutamate-bound model does not appear to fit the density very well. I would encourage the authors to supply a better/larger density figure in this panel so that the fit is clear.

Figure 5

I would encourage the authors to cross-check their references. In the legend of Figure 5, PDB:7cmi is listed as the legend "L-Arginine bound b0+AT (PDB:7cmi) (c)" but PDB:7cmi is the leucine bound b0+AT.

Page 5 "The repositioning of ectodomain results in a wider entrance vestibule, no doubt to accommodate the larger cystine ligand". I would rethink this sentence. There is, as I see it, no

evidence that the position of the ectodomain relates to the specific function or selectivity of the proteins. The difference could just as easily reflect a degree of conformational flexibility at the loose interface between the light and heavy chain portions of xCT.

Page 6 In my opinion, talking about an “extended” 6b helix is ambiguous. I would suggest “unwound” (or a similar term) to describe the change in secondary structure more accurately.

Page 7 “ Consistent with previously characterized members of the SLC7 family...”

Both the Apo and Glutamate structures have relatively high clash scores (and Molprobit scores). Also, the PDB reports indicate a poor fit to the maps around some critical areas (for example, residue R396 in the glutamate bound structure). I would strongly encourage the authors to improve the coordinates, especially in the areas around the binding site and entrance tunnel.

Reviewer #3 (Remarks to the Author):

The authors present cryo-EM structures of the glutamate/cysteine exchanger xc- in apo and glutamate and cystine-bound states. The structures are around 3.4 Å resolution and all in an inward facing conformation. As expected from other SLC7 structures, the transport domain (xCT) has the same architecture as the APC (LeuT-like) superfamily of transporters. Based on the structures, functional measurements, docking, and molecular dynamics simulations, the authors suggest an interesting new mechanism for fairly specific molecular recognition of glutamate and cystine. In particular, unwinding of part of helix 4 makes space for cystine while folding of this section shrinks the binding site for glutamate. Additionally, arginine residues coordinate the negatively charged carboxylates of the substrates.

Overall, this is a well written and convincing paper that clearly benefits from the synergy between experimental and computational approaches. I have no major issues with the paper and only a number of smaller comments and suggestions.

By the way, I liked the clear figures throughout, in particular the consistent color coding. The swarm plots with box plots for the transport assay data are also nice and comprehensive.

MINOR COMMENTS

1) p5 "hydrophobic cystine"

"Hydrophobic" might be a bit too strong; compared to the substrate of LAT1/2, cystine is still water soluble. The calculated xlogP is -6.3 (see pubchem CID 595) so cystine cannot really be called "hydrophobic" on an absolute scale.

2) p7 The import role of R396 is shown both through functional measurements of mutants (Fig 2b) and calculations (Fig 4).

The fact that R396A does not transport makes sense but can the authors speculate why R396K does not transport?

3) p8 The density of Glu (eg Fig 3b) seems fairly featureless and different orientations seem possible. Make clear how the orientation in the model was decided and what evidence the authors have to support their chosen orientation.

This question is of some relevance because cystine has two carboxylates whereas Glu only has one, so naively one could think that either orientation of Glu might fit. What is the evidence from MD simulations as to how Glu binds?

4) Similar to R396K, why does R135K does not transport? Would Cystine not interact with R135K?

5) p11 "Our results suggest that Arg396 contributes ~ 2 kcal/mol more free energy than Arg135"

Quote the actual mean free energy difference between R396A and R135A including the error. If one were just to look at table 4c one would think that in the row labeled 'Run 1', R135 contributes more, even though it makes no sense to assume that one had to compare just across rows. By calculating the difference properly, any kind of confusion would be avoided and the reader would also get a better sense of the accuracy.

The MMPBSA approach for the absolute binding free energy (ABFE) calculations is a relatively quick and computationally cheap way to estimate ABFEs; as some of the authors note in Ref 77: "[...] yet there is also value in MMPBSA calculations [...] if agreement to experimental affinities in absolute terms is not of interest." The limitations of the approach should be acknowledged somewhere. Additionally, " $\Delta\Delta G$ " between the quoted ABFEs in table 4c are probably more meaningful than the absolute values.

6) Suppl Fig 9

Figure is hard to read, increase fonts on subfigures (and zooming in does not work well because they appear as rastered images with limited resolution).

In addition to the RMSF for residues 330 - 339 also show the rest of the protein for comparison. Plot with an error estimate on the RMSF (from multiple repeats, block analysis, or boot strapping) to indicate if the difference between apo and substrate bound simulations is significant.

7) Fig 4

It would be useful to include a "ligand plot" (binding interactions) for glutamate similar to 4b for cystine, either in Fig 4 or supplementary information as this would make it easier to compare cystine and glutamate.

8) p 14 "In xCT it is likely a similar mechanism is at play"

Fix grammar: "In xCT is likely a similar mechanism at play"

REVIEWER COMMENTS

Reviewer #1 (Remarks to the Author):

The manuscript by Parker et al. presents cryo-EM structures of apo and glutamate-bound states of human xc- transporter, accompanied by functional analysis of WT and mutants, as well as in silico analysis of the cysteine-bound state. Despite several structural reports on SLC7 transport systems, this manuscript uncovers important details of the xc- transport mechanism, relevant in human health and disease, notably the first substrate-bound xc- structure, and a major improvement in resolution compared to a previous report. As such, I found this study of great interest for the membrane transport field, and recommend publication. The manuscript is clearly written, and the experimental design, as well as structural/functional analyses are sound. The comments below aim at improving clarity, or correcting mistakes:

1-Regarding sample preparation: in the first "results" section, it is mentioned that protein expression/purification, as well cryoEM sample preparation went through several rounds of optimisation. The "methods" section describes protocols in detail but there is no mention of optimisation strategies. It would be useful to know what was optimised, and what seemed to be important, as previous attempts to purify an xc-transporter used a different strategy based on consensus mutagenesis, and led to a lower resolution cryoEM structure. Also, since xc- is a cystine/glutamate exchanger one would have expected glutamate- and cystine-bound states, respectively, as targets for structural studies. A comment on why the cystine-bound complex was not pursued, or failed to yield high-resolution structures would again be informative.

We have added this information to the methods section, we found that the passage number of the cells was an important feature of the quality of the protein with passage numbers less than 10 required for high yield (>0.4mg/L), biophysically stable protein. We trialed a few different detergents with only a LMNG:CHS purified sample yielding a structure from the cryo-EM grids. As to a structure with cystine bound; cystine has very limited solubility in water and requires a high pH (1M HCL) to reach mM in solution. The low pH resulted in aggregation of the protein sample, and although we tried to get the complex using lower concentrations of cystine, this was ultimately unsuccessful. In comparison with the previous study (Oda K. et al. Protein Science 2020) the use of DDM:CHS and GDN appears to have resulted in less stable protein. This is also something we observed. The use of LMNG:CHS in this example appears to be the main reason for our success in working with the WT protein.

2-Regarding ectodomain/SLC7A11 interface, there is a mistake in Fig.1d, where instead of Q217 and Q219, it reads R135 twice. More importantly, at the end of page 5 the authors mentioned that a 20-degree opening compared to LAT1 “results in a wider vestibule, no doubt to accommodate the larger cystine ligand. SLC7A11 also has a noticeably less charged tunnel leading to the extracellular gate compared to LAT1 and LAT2, which may also facilitate binding of the hydrophobic cystine ligand.” I think these statements should be explained in more depth. The vestibule in published LAT1 structures is wide enough to let cystine enter, although it is not a substrate, so why the authors think that further widening of the vestibule is specifically associated to cystine binding? How does the vestibule seen in their structure compare to the published ~6 angstrom xc- structure? Could it be that the ectodomain is dynamic relative to the transporter? I would expect so, since they are linked through a disulfide bond, and likely weak contacts. Have the authors explored cryoEM 3D-variability or multi-body analyses to look into conformational heterogeneity?

We did explore the structural variability of the ECD relative to the transport domain. There is some tilting of ECD towards the extracellular loops of the transporter based on 3D variability analyses in cryosparc, but we see no improvement in ectodomain density throughout these analyses. This suggests, perhaps unsurprisingly that the ectodomain is more mobile than the other parts of the structure and that distinct conformations of the ECD with respect to the transporter are not reflected in our data – more like a continuum of motion.

With regards to our comment on the wider entrance vestibule, we have replaced ‘no doubt’ with ‘possibly’, to reflect this statement is currently conjecture. The larger vestibule in xCT is clearly present, although whether this plays a role in cystine acquisition is currently unknown. Nevertheless, cystine is a much larger ligand than single amino acids and is equivalent in size to a di-peptide. We think it therefore logical that in xCT the entrance vestibule would be larger to facilitate efficient uptake and reduce steric clashes during substrate capture.

Finally, in the section dealing with cystine binding, the authors mention that physiologically relevant cystine bears a formal negative charge, plus several polar alpha-amino and -carboxylate groups. Why is it described here as “hydrophobic cystine ligand”?

We have corrected our potentially misleading statement regarding the hydrophobicity of cystine. Where appropriate we have included the phrase ‘low water solubility’.

3-I found the graphical part of the manuscript a bit frugal, and some efforts could be put towards improving this part. For example: in figures throughout the main text the limits of the membrane are not included, and transporter orientations (Figs. 2-4) leave the impression that the bottom of the extracellular vestibule reaches halfway across hydrophobic core of the membrane.

We thank the referee for pointing this out, we have now further illustrated the figures with membrane limits and distances.

Cryo-EM maps contain clear density for the micelle and the authors could use them to estimate membrane "borders". Also, one of the main differences observed in xc- apo structure compared to previous SLC7 structures is the extended conformation of TM6b, but this is shown in a supplementary Fig (Supp Fig 5). The same applies to K198 position. Why not to highlight these structural aspects in main-text figures?

Again, we thank the referee for this suggestion. We have now moved these figures to the main text.

4-There is a typo at the end of page 7 "Consistent with previously members of the SLC7 family,".

Thank you, now corrected

Reviewer #2 (Remarks to the Author):

Parker et al present a structural study of the human xCT cystine/glutamate antiporter which suggests a mechanism for ligand recognition and transport. There are a few issues that I would like addressed before publication.

Q1. I am curious to learn why the authors have not attempted to capture the antiporter in the presence of cystine, instead of relying solely on molecular dynamics.

As discussed above in response to a comment from referee 1, we did attempt to capture this complex, but our efforts were unsuccessful. However, given the current state of the transporter in the inward

open conformation we think the current structure provides valuable insight into the process of glutamate binding from the cytoplasmic side. We feel it would be more informative to focus future efforts on trapping the outward facing state bound to cystine to understand more clearly the mechanism of cystine recognition from the extracellular side of the membrane.

Q2. There appear to be several novel secondary structure transitions that occur in xCT during transport. The authors mention the conformation of TM6B, but this is not explored further so it is never confirmed if this feature is relevant in protein function. Given that this appears to be one of the unique structural features of the human xCT, I think the manuscript would be strengthened by a closer look at the importance of this segment in its function.

The referee is correct in highlighting this aspect of the structure. Indeed, TM6B displays a very interesting difference compared to other SLC7 structures reported to date. We have gone back to our molecular dynamics analysis of the structure and analysed the root mean square fluctuation of this region more closely. Our analysis, shown in SI Fig 5. reveals that the unwound structure is stable in the simulations, lending further strength to our observations and modelling. How to address the impact of this feature on transport? This is a very tricky question as dead variants of residues in this region would not provide much insight. Indeed, mutations of the conserved tyrosine on TM6B in LAT1 resulted in transport defective protein. Therefore, to dissect out the reason and importance of this region, we feel that alternative conformations of the transporter are going to be the most informative. We are working on this strategy with nanobody binders. However, we feel this level of additional work is not warranted in this study, as we feel the current structural and functional insights are sufficiently robust to stand alone at this stage. We hope the referee agrees.

Q3. In general, I would like to see the main figures relate better to the manuscript text. Specifically, I would like the authors to make the following changes:

Figure 1

- Include a panel to show a better representation of the protein topology. Despite this being a common fold, the readers would benefit from a good illustration.

We have added a colour coded topology panel to this figure

- Panel 1 d- the residues are mislabelled.

This has been corrected

Figure 2

- I would strongly recommend adding panels to illustrate the structure described in the text: "The binding site is flanked on one side by TM1 and TM6, which are broken in the centre of the transporter to form two discontinuous helices, termed 1A, 1B and 6A, 6B respectively. At the opposite side of the binding site, TMs 3, 4, 8 & 9 form the hash motif 38, against which TMs 1 and 6 move during alternating access transport."

And

"The extracellular side of the binding site is closed through the close packing of TM1B and 6A against TM4 and 10, with a strictly conserved aromatic side chain, Tyr244 on TM6A, sealing the binding site entrance. A conserved and essential lysine on TM5, Lys198, which in previous SLC7 family structures acts to coordinate interactions between the unwound region of TM1 and TM8 does not interact with TM8 in xCT"

We have now moved ED Fig. 5 to Fig. 2b and included a new panel to show the location of Lys198. We have also added labels to Fig. 2a to more clearly identify the regions described in the text.

Figure 3

In panel 3c the glutamate-bound model does not appear to fit the density very well. I would encourage the authors to supply a better/larger density figure in this panel so that the fit is clear.

We do show the density for each TM helix in the cryo-EM workflow figures (Figs. S2 and S7), which show the model does fit the density well. However, the Glu-bound model is of a lower resolution to the apo structure, which will impact on the features in the maps. Nevertheless, the side chains do sit in their respective densities and this map is contoured to exemplify the main features of the helix for the figure. At lower contours the smaller side chains do come through, but with more noise in other parts of the map. The important point for us in this figure was to show in the Glu-bound structure TM8 is clearly helical, which it clearly isn't in the apo structure. For this reason, we would like to keep this figure as originally drawn, and hopefully readers can view the maps in coot or chimera X for regions they are particularly interested in.

Figure 5

I would encourage the authors to cross-check their references. In the legend of Figure 5, PDB:7cmi is listed as the legend "L-Arginine bound bo+AT (PDB:7cmi) (c)" but PDB:7cmi is the leucine bound bo+AT.

Many thanks, these mistakes have been corrected.

Page 5 "The repositioning of ectodomain results in a wider entrance vestibule, no doubt to accommodate the larger cystine ligand". I would rethink this sentence. There is, as I see it, no evidence that the position of the ectodomain relates to the specific function or selectivity of the proteins. The difference could just as easily reflect a degree of conformational flexibility at the loose interface between the light and heavy chain portions of xCT.

Agreed. We have replaced 'no doubt' with 'possibly' to reflect this statement as being one of conjecture on our part.

Page 6 In my opinion, talking about an "extended" 6b helix is ambiguous. I would suggest "unwound" (or a similar term) to describe the change in secondary structure more accurately.

Agree, we have amended the text accordingly.

Page 7 " Consistent with previously characterized members of the SLC7 family..."

Corrected

Q4. Both the Apo and Glutamate structures have relatively high clash scores (and Molprobability scores). Also, the PDB reports indicate a poor fit to the maps around some critical areas (for example, residue R396 in the glutamate bound structure). I would strongly encourage the authors to improve the coordinates, especially in the areas around the binding site and entrance tunnel.

For SLC7A11 the apo model has a clash and Molprobability score of 15.39 and 2.28, and the glu-bound model has a clash and Molprobability score of 11.18 and 2.02, respectively. Despite the lower nominal

resolution of the glu-bound map compared to apo (3.7Å vs 3.4 Å), the glu-bound model has better validation statistics, likely due to excluding the ECD of SLC3A2.

We have again taken another look at our maps for both structures, however, we feel the current models best represent the maps we have. Indeed, the Molprobity scores are significantly lower than the nominal resolution of our maps, arguing that our models are appropriate for the resolution.

Nevertheless, we also compared our models to those in the literature for the validation statistics for human LAT1–4F2hc (Yan, R. et al., *Nature*, 2019):

Their model statistics are pretty much in line with ours (worse MolProbity, slightly better clashscore – their ECD density is likely stronger than ours).

Concerning the poor fit around certain side chains. There isn't all that much we can do about this, especially for the glu-bound model. It is a much lower resolution model/map compared to apo, and density for some side chains is therefore weaker. This could be attributed to fewer particles going into the reconstructions (there were over 4x the number of particles going into the apo map for example) or generally higher mobility due to the presence of the ligand triggering instability in line with conformational changes during transport.

Here's a comparison of R396 for apo (left panel), and R396 for glu-bound (middle and right panel, different contour levels)

The poor fit for R396 in the glu-bound map is because of the weaker density. There is still connecting density at lower contour level (0.1 rmsd).

Validation reports for EM maps are stringent when it comes to fits to density. So, it's not that the model is bad *per se*, it's that density is weaker for these side chains and the validation reports are flagging this.

Here are the residue-property plots for our glu-bound model:

- Molecule 2: Cystine/glutamate transporter

For comparison, the residue-property plots for LAT1:

Overall, this is a well written and convincing paper. I have no major issues with the paper and only a number of smaller comments and suggestions.

By the way, I liked the clear figures throughout, in particular the consistent color coding. The swarm plots with box plots for the transport assay data are also nice and comprehensive.

We thank the referee for this comment, and we hope the additions made in response to referee 1 and 2 above further improve these for the readers.

MINOR COMMENTS

1) p5 "hydrophobic cystine"

"Hydrophobic" might be a bit too strong; compared to the substrate of LAT_{1/2}, cystine is still water soluble. The calculated xlogP is -6.3 (see pubchem CID 595) so cystine cannot really be called "hydrophobic" on an absolute scale.

We have amended this in the revised text, thank you.

2) p7 The import role of R396 is shown both through functional measurements of mutants (Fig 2b) and calculations (Fig 4). The fact that R396A does not transport makes sense but can the authors speculate why R396K does not transport?

We assume that because this side chain cannot be replaced by lysine, that charge alone is unlikely to be the main role played by R396. Our current guess is that arginine provides a more stable interacting partner for both cystine and glutamate, as arginine is less flexible. Indeed, given the increased flexibility of TM6B and induced fit mechanism we propose for glutamate and cystine recognition, the structural rigidity of arginine on TM10 may be a key requirement for transport. We have added the following additional clarification statement on p.11 **Our functional assays confirm the necessity of arginine in this position (Fig. 2d), demonstrating that additional structural contributions, possibly involving the stabilisation of TM8 following ligand binding, are required in addition to its cationic properties.**

Additionally, transport is a multistep process for which our assay can only measure successful completion of the full cycle. Although our results confirm that a lysine cannot substitute for arginine for the full transport cycle, it doesn't inform on all previous steps. We are currently undertaking follow

up studies to investigate this question, although we feel these are outside the scope of the current study.

3) p8 The density of Glu (eg Fig 3b) seems fairly featureless and different orientations seem possible. Make clear how the orientation in the model was decided and what evidence the authors have to support their chosen orientation. This question is of some relevance because cystine has two carboxylates whereas Glu only has one, so naively one could think that either orientation of Glu might fit. What is the evidence from MD simulations as to how Glu binds?

SLC7A11 belongs to the wider APC superfamily, for which several structures in complex with amino acids exist (selected recent references ¹⁻⁴). In all these structures the amino acid is orientated so that the carboxy and amino groups interact with the discontinuous regions in TM1 and TM6 respectively, illustrated below for GkApcT, a bacterial homologue of SLC7A1 ¹. You can also see that even in a reasonably high-resolution structure of 2.9 Å (5oqt) structure, the density for the L-alanine amino acid is featureless, similar to the L-glutamate density in our cryo-EM maps. Thus, our modelling of the ligand in the current orientation draws on past observations from closely related members of the APC superfamily. We have now added the following sentence to explain why modelled the ligand this way around "In particular, the amino and carboxy groups of the ligand coordinate with free carboxy and amino groups in the discontinuous regions of TM1 and TM6 respectively, with a water molecule linking the carboxy group to TM7 ¹"

Figures of the bound amino acids in GkApcT (5oqt), taken from ¹.

4) Similar to R396K, why does R135K does not transport? Would Cystine not interact with R135K?

Similar to our reasoning for R396, our results would suggest that charge alone is not sufficient to enable transport. This points to a requirement for Arginine, again, possibly related to the structural properties of this side chain. However, we are currently unclear on what steps require an arginine specifically.

5) p11 "Our results suggest that Arg396 contributes ~ 2 kcal/mol more free energy than Arg135" Quote the actual mean free energy difference between R396A and R135A including the error. If one were just to look at table 4c one would think that in the row labeled 'Run 1', R135 contributes more, even though it makes no sense to assume that one had to compare just across rows. By calculating the difference properly, any kind of confusion would be avoided, and the reader would also get a better sense of the accuracy.

The MMPBSA approach for the absolute binding free energy (ABFE) calculations is a relatively quick and computationally cheap way to estimate ABFEs; as some of the authors note in Ref 77: "[...] yet there is also value in MMPBSA calculations [...] if agreement to experimental affinities in absolute terms is not of interest." The limitations of the approach should be acknowledged somewhere. Additionally, " $\Delta\Delta G$ " between the quoted ABFEs in table 4c are probably more meaningful than the absolute values.

We thank the referee for making this observation and we have amended the table to include $\Delta\Delta G$ values along with a detailed explanation of our statistics in the methods section. We have added a sentence in our methods section to indicate the potential limitations of the MMPBSA method as opposed to more rigorous FE methods.

6) Suppl Fig 9

Figure is hard to read, increase fonts on subfigures (and zooming in does not work well because they appear as rastered images with limited resolution). In addition to the RMSF for residues 330 - 339 also show the rest of the protein for comparison. Plot with an error estimate on the RMSF (from multiple repeats, block analysis, or boot strapping) to indicate if the difference between apo and substrate bound simulations is significant.

We have amended the image to increase the resolution. We have generated RMSF plots again and calculated the error bars with bootstrapping, as the reviewer recommends. We have plotted the RMSF for each residue as a thin line and the error bars for each point are reflected by the width of the line at each residue. As can be seen the error bars are very small after 1000 bootstrapped samples and they did not change with the number of bootstrapped samples.

7) Fig 4

It would be useful to include a "ligand plot" (binding interactions) for glutamate similar to 4b for cystine, either in Fig 4 or supplementary information as this would make it easier to compare cystine and glutamate.

We thank the referee for pointing out this omission. We have now included a ligand plot in ED Fig.8c.

8) p 14 "In xCT it is likely a similar mechanism is at play" Fix grammar: "In xCT is likely a similar mechanism at play"

Many thanks!

References

- 1 Jungnickel, K. E. J., Parker, J. L. & Newstead, S. Structural basis for amino acid transport by the CAT family of SLC7 transporters. *Nature communications* **9**, 550, doi:10.1038/s41467-018-03066-6 (2018).
- 2 Shaffer, P. L., Goehring, A., Shankaranarayanan, A. & Gouaux, E. Structure and mechanism of a Na⁺-independent amino acid transporter. *Science (New York, NY)* **325**, 1010-1014, doi:10.1126/science.1176088 (2009).
- 3 Errasti-Murugarren, E. *et al.* L amino acid transporter structure and molecular bases for the asymmetry of substrate interaction. *Nature communications* **10**, 1807, doi:10.1038/s41467-019-09837-z (2019).
- 4 Ilgu, H. *et al.* Insights into the molecular basis for substrate binding and specificity of the wild-type L-arginine/agmatine antiporter AdiC. *Proc Natl Acad Sci U S A* **113**, 10358-10363, doi:10.1073/pnas.1605442113 (2016).

REVIEWER COMMENTS

Reviewer #1 (Remarks to the Author):

The authors have fully addressed my concerns, and I don't have further points to raise

Reviewer #2 (Remarks to the Author):

The authors have addressed all of my concerns and queries in the revised manuscript.